# The Role of Smartphone Use in Sensory Processing: Differences Between Adolescents with ADHD and Typical Development

**DOI:** 10.3390/ijerph21121705

**Published:** 2024-12-21

**Authors:** Rosa Angela Fabio, Rossella Suriano

**Affiliations:** Department of Cognitive, Psychological and Pedagogical Sciences and Cultural Studies, University of Messina, 98100 Messina, Italy; rossella.suriano@studenti.unime.it

**Keywords:** smartphone use, sensory processing, attention deficit hyperactivity disorder

## Abstract

The use of smartphones is widespread among adolescents and can affect various cognitive processes. However, the effects of smartphone use on sensory processing, particularly among individuals with attention deficit hyperactivity disorder (ADHD), remain largely unknown. The present study investigated the relationship between smartphone use intensity and sensory processing in adolescents with typical development and those with ADHD. The sample included 184 adolescents aged 14 to 18 years (M = 16.56; SD = ±1.87), with 92 diagnosed with ADHD and 92 with typical development, matched for age, gender, and IQ. Participants completed a self-report questionnaire to measure smartphone use intensity, while sensory processing was assessed using the Adolescent Sensory Profile (ASP). The results revealed a significant association between the intensity of smartphone use and heightened sensory responses in adolescents with typical development. However, this relationship was not observed in participants with ADHD. These preliminary findings suggest that smartphone use may influence sensory processing differently depending on neurotypical development or the presence of ADHD, potentially contributing to the promotion or mitigation of sensory dysfunctions. Future studies are needed to further explore the mechanisms underlying these differences and to better understand the impact of digital technologies on sensory functioning.

## 1. Introduction

The daily lives of individuals are increasingly shaped using technological devices [1,2,3]. Among these, the smartphone is arguably the most widespread worldwide due to the numerous advantages it offers in terms of accessibility, convenience, and multi-functionality [4,5,6]. Its ability to combine communication, entertainment, information, and productivity into a single tool makes it extremely versatile and indispensable in daily life [7,8,9]. Although smartphone use provides numerous benefits, several studies have highlighted how excessive use is associated with a range of negative consequences on cognitive functioning [10,11,12]. Among these, a significant reduction in attention and concentration capacity is observed, as well as an increase in cognitive dependence for memory and information retrieval [13,14]. Prolonged screen exposure is correlated with a decline in sleep quality, which in turn impairs cognitive function [15,16]. Additionally, intensive device use is associated with impaired executive functioning, reduced ability to process complex information, and difficulties in problem-solving [17,18,19,20]. Despite the extensive debate on the cognitive effects of smartphone use, studies on the impact of these devices on sensory processing remain limited.

Sensory processing is the process by which the nervous system receives, interprets, and organizes information from the various senses [21,22,23]. This process allows the brain to transform sensory data into useful responses, enabling an individual to interpret and appropriately react to the surrounding environment, thereby influencing adaptation and overall well-being [24,25]. Dunn’s model [26] describes the variability in sensory processing responsiveness through two continua: one related to the neurological threshold and the other to behavioral self-regulation responses. The neurological threshold can be low (hyper-reactivity) or high (hypo-reactivity), while the behavioral responses can be passive (acceptance of stimuli) or active (management of stimuli). The interaction between these continua generates four sensory processing components: sensation seeking, sensory avoidance, sensory sensitivity, and low registration.

Sensation seeking refers to individuals with a high sensory threshold who seek intense experiences to stimulate their senses. Adolescents with this profile might become distracted by background noise, such as the hum of a fan or the sound of typing, while studying.

Sensory avoidance pertains to those who, despite having a low threshold, avoid intense stimuli to prevent overload. An adolescent who is sensitive to sensory overload might avoid crowded or noisy spaces to maintain comfort.

Sensory sensitivity refers to individuals with a low threshold who perceive even minimal stimuli intensely. Adolescents with this profile might become distracted by background noise, such as the hum of a fan or the sound of typing, while studying.

Finally, low registration involves individuals who do not readily notice stimuli, showing disinterest or distraction. An individual in this category may not notice important sensory stimuli, such as a ringing phone or people talking, even though these stimuli are noticeable to others (Table 1).

These components influence the perception and reaction to different sensory modalities, including taste, smell, movement, vision, touch, and hearing [27].

Smartphones simultaneously stimulate multiple sensory channels, yet little is known about how this type of stimulation may alter sensory processing and impact adaptive responses, especially in young adolescents [28,29,30]. Some studies have highlighted how early exposure to digital media may interfere with sensory development, suggesting potential long-term effects [30,31,32]. Dadson et al. [33] emphasized the relationship between screen time and atypical sensory processing abilities in children. Tekeci et al. [34] underscored the importance of reducing screen time, including smartphone use, to prevent negative outcomes. Only Park and Chang [28] and Hong and Lee [29] have studied smartphone use in university students, finding it to be associated with difficulties in sensory processing, suggesting that intensive use of digital devices may also affect these abilities in younger individuals.

Furthermore, we are unaware of any studies that have investigated this relationship in a sample of individuals with neurocognitive disorders, such as attention deficit hyperactivity disorder (ADHD). ADHD is a clinical condition characterized by persistent and pervasive patterns of inattention, hyperactivity, and impulsivity, which can be categorized into three subtypes based on symptomatology: predominantly inattentive (ADHD-I), predominantly hyperactive/impulsive (ADHD-H), or combined (ADHD-C) [35]. As widely demonstrated in the literature, sensory component parameters are significantly higher in individuals with ADHD [36,37,38,39]. Moreover, several studies have highlighted that smartphone use is greater in individuals with ADHD compared to those with typical development [40,41,42,43]. However, the factors that may contribute to high smartphone use in ADHD are unclear. One possible explanation may stem from the sensory modulation theory [44], which conceptualizes digital devices as tools for sensory self-regulation. According to this theory, media devices are used to create sensory balance as follows: on one hand, they effectively stimulate the sensory system, and on the other, they offer the ability to modulate the intensity of sensory input, allowing individuals to manage their perceptual experiences based on the context. This concept is particularly relevant for individuals with ADHD, who often seek more intense and engaging sensory experiences to compensate for the sensory regulation difficulties typical of their condition [45,46,47]. In the case of smartphone use, the device could offer adaptive solutions capable of meeting the diverse perceptual needs of individuals with ADHD.

The present study aims to explore how smartphone use may be associated with sensory processing in response to stimuli, examining potential differences and similarities in the impact of digital habits among adolescents with different developmental profiles. More specifically, the goal is to investigate the relationship between smartphone usage intensity and sensory processing in adolescents with typical development and those with ADHD. Based on the existing literature, it is expected that (a) adolescents with ADHD will exhibit higher values in the sensory component parameters compared to their typically developing peers; (b) smartphone use will be more intense in adolescents with ADHD than in those with typical development; and (c) among typically developing adolescents, higher smartphone use will be associated with higher values in sensory component parameters, while this relationship remains unclear for adolescents with ADHD. Considering sensory modulation theory [44], it is hypothesized that smartphone use may not have negative effects on sensory processing in individuals with ADHD, suggesting a potential regulatory function of digital device use within this population. In more detail, for individuals with a high sensory threshold, who tend to seek intense stimulation, the smartphone could provide visual, auditory, and tactile stimuli, helping maintain attention and motivation. For those with a low sensory threshold, the device could allow for the modulation of stimulus intensity, such as adjusting screen brightness or volume, thereby preventing sensory overload. Individuals with tendencies toward sensory avoidance might use the smartphone as a self-regulation tool, selecting more controllable and less intrusive stimuli. Finally, for those with low sensory registration, the notifications and vibrations of the device could stimulate attention, enhancing awareness and sensory engagement.

## 2. Materials and Methods

### 2.1. Participants

The participants were selected from an initial sample of 1421 adolescents, aged between 14 and 18 years (M = 16.58; SD = ±1.86), including 820 males and 601 females. All participants were Italian and attended public secondary schools in Sicily, a region in southern Italy. Each participant completed the Diagnostic Assessments by filling out the Adult Self-Report Scale (ASRS), which identified 104 participants with scores consistent with ADHD. Of these, 92 were subsequently confirmed as cases of ADHD after a thorough clinical evaluation through a psychological interview. Other potentially conflicting conditions were excluded based on normal scores on the ASRS (0 to 17), normal scores on the Raven Progressive Matrices, and a clinical interview. None of the participants had a history of brain injuries, epilepsy, psychosis, or anxiety disorders.

The control group was selected from the initial sample of 1421 participants who had obtained normal scores on the ASRS (0 to 17) and had no clinical disorders, as confirmed by a clinical interview. It consisted of 92 subjects matched to the ADHD group in terms of gender, age, and IQ, following the guidelines of Kover and Atwood [48], to ensure an adequate match between the two groups.

Descriptive statistics for each variable characterizing the final sample, consisting of 92 subjects with ADHD and 92 subjects with typical development, are presented in Table 2. Each group consisted of 57 males and 35 females; thus, there were no significant gender differences. Similarly, there were no significant differences in age (*t* = 0.85, *p* = 0.12) and intelligence (*t* = 0.67, *p* = 0.33) between the ADHD group and the typical development group. Symptoms of inattention and hyperactivity were significantly more prevalent in the ADHD group compared to the control group (*t* = 36.01, *p* < 0.001; *t* = 31.12, *p* < 0.001).

### 2.2. Procedure

The study was conducted in accordance with the principles of the Declaration of Helsinki. Each participant voluntarily agreed to participate, and informed consent was obtained from the parents or guardians of minor participants. Participants aged 18 or older provided their own consent before the study commenced. After completing the initial assessment phase, self-report questionnaires were administered. Specifically, the Adolescent Sensory Profile (ASP) was used to examine the sensory experiences of the adolescents, and the smartphone use questionnaire was used to assess the intensity of smartphone usage. All tests were administered in a quiet room during school hours in the morning to ensure standardized conditions and minimize variables that could influence the results.

### 2.3. Measurement

#### 2.3.1. Diagnostic Assessments

For the assessment of symptoms of inattention, hyperactivity, and impulsivity related to ADHD, in accordance with the diagnostic criteria of the DSM-V [35], the Adult Self-Report Scale [49,50,51] was used. This scale consists of 18 items evaluates the core symptoms of ADHD, which are divided into two dimensions: inattention and hyperactivity/impulsivity. Responses are given on a 5-point Likert scale, ranging from 0 (never) to 4 (very often). The ASRS has demonstrated good psychometric properties, with validity and reliability supported by studies conducted both on clinical samples and general populations [52,53]. The internal consistency, measured by Cronbach’s alpha coefficient, showed a value of α = 0.88 [54]. Furthermore, the Italian version of the scale was translated and validated by Somma et al. [55]. In addition, Adler et al. [56] confirmed that the symptom checklists used in the ASRS are internally consistent self-assessment tools for the diagnosis of ADHD even in adolescence.

For the assessment of cognitive abilities, Raven’s progressive matrices [57,58] were used. This tool is widely used in psychology, education, and professional settings to measure fluid intelligence, which represents the ability for abstract reasoning, problem-solving, and adapting to new situations. It is based on a series of incomplete visual patterns, where the task is to complete each pattern by selecting from the proposed options the one that correctly completes the logical sequence. The standard version contains a sequence of 60 problems, divided into 5 sets (A, B, C, D, E), with increasing difficulty.

To confirm the diagnosis of ADHD, exclude the presence of other disorders in the ADHD group, and verify the absence of clinical conditions in the control group, a clinical interview was conducted based on the DSM-5 guidelines [35]. The interview, lasting approximately one hour, explored various aspects related to general health, daily functioning, and academic performance, as well as difficulties associated with ADHD [59,60,61]. Participants were encouraged to express themselves freely in their own words, and when necessary, prompts were used to elicit more detailed responses. Behavioral observations during the interview contributed to the assessment of symptoms and supported the formulation of the diagnoses.

#### 2.3.2. Smartphone Use Questionnaire

To assess the intensity of smartphone use, a questionnaire consisting of 6 items was adapted from Lee et al. [62]. Participants were asked to respond by indicating the frequency with which certain behaviors occurred, using a 4-point Likert scale. To ensure the reliability of the questionnaire, an internal consistency analysis was conducted, which yielded a Cronbach’s α value of 0.87, indicating good consistency among the items. Table 3 presents the full smartphone use questionnaire, including the item descriptions and response options.

#### 2.3.3. Adolescent Sensory Profile

The Adolescent Sensory Profile (ASP) is a diagnostic tool based on Dunn’s [26] sensory processing model, used to assess how individuals perceive, interpret, and respond to sensory stimuli from their environment. The model includes four sensory components, which result from the interaction between sensory thresholds and reactivity to stimuli, arranged along a continuum [63]. The extreme points of each scale give rise to four distinct quadrants. Specifically, the low registration quadrant represents a high threshold accompanied by a passive response, while sensation seeking corresponds to a high threshold with an active response to stimuli. In contrast, sensory sensitivity is associated with low thresholds, with a passive response, while sensation avoiding is the quadrant representing low thresholds with an active response. Specifically, sensory avoiding and sensation seeking are distinguished by their active responses to stimuli; while the former focuses on avoiding sensory overload, the latter involves actively seeking out intense sensory experiences. The sensory profile also includes six fundamental sensory modalities that cover various aspects of sensory experience: taste and smell, movement, vision, touch, activity level, and auditory processing. Together, the components and modalities provide a comprehensive overview of the different sensory responses. The questionnaire consists of 60 items, with a 5-point Likert scale response format (from 1 = almost never to 5 = almost always). In the current study, the Cronbach’s Alpha values calculated for each sensory component and modality of the questionnaire reflect good internal consistency. For the sensory components, the alpha coefficients were 0.81 for low registration, 0.78 for sensation seeking, 0.84 for sensory sensitivity, and 0.89 for sensation avoiding.

### 2.4. Statistical Analysis

The statistical analysis was conducted using SPSS software version 26.0 (SPSS Inc., Chicago, IL, USA). The measured parameters included sensory profile components (low registration, sensation seeking, sensory sensitivity, and sensation avoiding) and the intensity of smartphone use across each experimental condition (ADHD participants and typically developing participants).

To address the first hypothesis, a 2 (Group: ADHD, typical development) x 4 (sensory quadrant: low registration, sensation seeking, sensory sensitivity, sensation avoiding) repeated-measures analysis of variance (ANOVA) was performed to evaluate differences in sensory processing profiles between adolescents with ADHD and those with typical development. Sensory quadrant scores were treated as the within-subjects factor, while group membership served as the between-subjects factor. Mauchly’s test confirmed that the assumption of sphericity was met, *χ^2^*(5) = 4.89, *p* > 0.05. For the second hypothesis, an independent-samples t-test was conducted to compare the smartphone use intensity between the ADHD and typical development groups. To address the third hypothesis, Pearson correlation analyses were conducted separately for the typical development and ADHD groups to examine the relationship between smartphone use and the different components of sensory processing dysfunction. For all tests, the significance level was set at *p* < 0.05.

## 3. Results

Table 4 presents the means and standard deviations for each of the sensory profile components in the two groups of participants with ADHD and typical development.

The first hypothesis predicted differences in the sensory profile quadrants between the ADHD and typical development groups. The analysis revealed a significant main effect of group, *F*(1, 182) = 27.12, *p* < 0.001, *η*^2^ = 0.13, indicating that participants with ADHD exhibited significantly higher levels across all sensory quadrants compared to the typical development group. A significant main effect of the sensory quadrant was also observed, *F*(3, 546) = 22.43, *p* < 0.001, *η*^2^ = 0.11, suggesting variability among the different quadrants. Specifically, the first sensory component, low registration, showed a significantly lower mean compared to the other quadrants (*t* = 2.88, *p* < 0.05; *t* = 2.31, *p* < 0.001; *t* = 2.95, *p* < 0.01). This means that both ADHD and TD participants scored lower in low registration and exhibited a reduced sensitivity to sensory stimuli compared to the other quadrants (sensation avoiding, sensation seeking, and sensory sensitivity). Furthermore, no significant interaction was found between group and sensory quadrant, *F*(3, 546) = 1.07, *p* = 0.36, *η*^2^ = 0.09, indicating that participants with ADHD consistently scored higher across the various quadrants. Post hoc analyses using Bonferroni correction confirmed that adolescents with ADHD scored significantly higher in low registration, sensory sensitivity, sensation seeking, and sensation avoiding compared to the typical development group (*ps* < 0.0125).

Regarding Hypothesis 2, which posited differences in smartphone use intensity between the ADHD and typical development groups, the results of the independent-samples t-test indicated a significant difference, *t* (182) = 12.3, *p* < 0.001, *d* = 0.78. Adolescents with ADHD (M = 15.95, SD = 3.52) showed significantly more smartphone use intensity compared to the typical development group (M = 10.88, SD = 2.88).

With reference to the correlation between smartphone use intensity and difficulty in managing sensory stimuli, for the typical development group, the results showed a significant positive, as evidenced by the significant correlations between smartphone use and sensory processing across all sensory components: *r* = 0.370, *p* < 0.01; *r* = 0.340, *p* < 0.01; *r* = 0.443, *p* < 0.001; *r* = 0.325, *p* < 0.01 (Figure 1). This suggests that greater smartphone use is associated with increased difficulties in sensory processing, indicating that adolescents with typical development may experience sensory overload due to intense device interaction.

In the ADHD group, no significant correlation was found between smartphone use intensity and sensory processing across all sensory components: r = 0.055, *p* = 0.65; r = −0.230, *p* = 0.06; r = −0.099, *p* = 0.42; r = −0.109, *p* = 0.37 (Figure 2). This suggests that smartphone use does not have a significant impact on sensory processing in adolescents with ADHD, indicating that there may not be a direct relationship between device use and sensory difficulties in this population. It is possible that for adolescents with ADHD, smartphones may serve as a self-regulation tool, helping to modulate sensory input rather than causing overload, as seen in the typical development group.

## 4. Discussion

The present study explored the relationship between smartphone usage intensity and sensory processing in adolescents with typical development and those with ADHD. The first objective was to examine differences in sensory profiles between the two groups, and the results revealed that adolescents with ADHD scored significantly higher in all components of the sensory profile (low registration, sensory sensitivity, sensation seeking, and sensation avoiding) compared to their typically developing peers. These findings were expected and are consistent with the existing literature, which describes individuals with ADHD as having greater difficulty in sensory processing [64,65,66]. According to Dunn’s sensory processing model [26], individuals with ADHD tend to exhibit more intense and less adaptive responses to sensory stimuli, likely due to a reduced ability to filter sensory information and a less efficient self-regulation system [67,68,69].

The second hypothesis of the study concerned the intensity of smartphone use in the two groups. The results obtained support the notion that adolescents with ADHD make more intensive use of the smartphone compared to their typically developing peers. This finding aligns with several studies linking ADHD to greater smartphone dependence, likely associated with characteristics such as stimulus-seeking behavior, difficulties in regulating impulsivity, and a tendency to prefer activities that offer rapid and immediate gratification [70,71,72].

The most innovative aspect of this study lies in the analysis of differences in the relationship between smartphone use and sensory difficulties across the two groups. Among typically developing individuals, a positive correlation was observed between the intensity of smartphone use and sensory difficulties, suggesting that prolonged smartphone use may contribute to sensory overload, resulting in challenges in stimulus regulation. Prolonged exposure to digital stimuli may interfere with sensory self-regulation processes, leading to less adaptive responses to stimuli in the physical environment [73,74,75]. These findings align with concerns raised in recent studies linking intensive smartphone use to issues of sensory dysregulation and reduced capacity to process external stimuli [76,77]. In contrast, among participants with ADHD, no statistically significant correlation was found between smartphone use intensity and sensory difficulties. This finding may suggest that the use of digital devices does not exacerbate pre-existing sensory issues in these individuals, but rather may serve as a self-regulation tool, allowing them to achieve sensory balance through a controllable and predictable medium [78,79]. The sensory modulation theory proposed by Harrison et al. [44] provides an explanation for this dynamic, suggesting that adolescents with ADHD may use smartphones as a kind of safe space that enables them to modulate their sensory experiences within a context they perceive as comfortable and under control.

Moreover, it is possible that in individuals without ADHD, smartphone use may assist with self-regulation, particularly among those who already exhibit higher sensory profile scores. These individuals may have heightened sensory sensitivity or a lower threshold for sensory input, making them more prone to sensory overload. In such cases, for them as well, smartphones could provide a means of controlling or moderating sensory experiences. For example, adjusting screen brightness, controlling notifications, or choosing specific apps and content could help these individuals manage the intensity of stimuli they encounter throughout the day. Thus, while excessive smartphone use may exacerbate sensory difficulties for some typical development individual, for others, it may function as a tool for regulating sensory input, potentially helping to maintain focus, prevent overstimulation, and facilitate emotional balance. This suggests that smartphone use may serve as a compensatory mechanism, helping individuals with higher sensory sensitivity to better navigate their sensory environment rather than contributing to further sensory overload.

This study represents the first attempt to compare the association between smartphone use and sensory processing in adolescents with typical development and those with ADHD. It contributes to the existing literature by highlighting the importance of considering the different impacts of digital device use according to the sensory profiles and self-regulation strategies characteristic of each group.

This study has several limitations, including the cross-sectional nature of its design, which prevents drawing causal conclusions about the effect of smartphone use on sensory processing and limits the ability to observe changes over time or explore causal dynamics. Another important limitation concerns the characteristics of the sample, which consisted exclusively of Italian adolescents attending public secondary schools in Sicily. While this provided a homogeneous group for the initial exploration of our research questions, it restricts the generalizability of the findings to adolescents from other cultural, socio-economic, or geographical contexts. Future research should aim to replicate these findings using more diverse samples, including participants from different regions, cultural backgrounds, and socio-economic conditions, to enhance the representativeness and applicability of the results.

The measurement of smartphone use intensity via self-reporting is another critical point. Although the questionnaire used was adapted from validated measures in the literature and demonstrated good internal consistency (Cronbach’s α = 0.87), self-reported data may be subject to response bias, potentially affecting the accuracy of the results. Future research should aim to integrate objective tools, such as passive monitoring apps, which could provide a more precise and reliable assessment of smartphone usage in terms of time and frequency. These tools should also be further validated within ADHD populations to enhance their applicability and utility in this area of research. Moreover, this study also did not differentiate between specific types of smartphone usage, such as usage for academic purposes (e.g., homework) versus recreational activities (e.g., entertainment). It is possible that these different types of usage have distinct impacts on sensory processing in adolescents with ADHD. Future research should investigate this variable to provide a more nuanced understanding of the relationships between smartphone use and sensory processing, allowing for tailored interventions and recommendations.

In future research, it would be valuable to replicate these findings with a larger sample and include different age groups to assess the generalizability of the observed associations between smartphone use and sensory processing in adolescents with ADHD compared to their typically developing peers. This study highlights interesting correlations based on essentially quantitative data. However, it would be valuable to examine the types of smartphone use; say, for example, an adolescent using it for homework versus one using it for online gaming. Further longitudinal studies could provide a deeper understanding of the temporal dynamics between smartphone use and sensory responses, allowing for the assessment of causal effects and the direction of observed associations. Finally, the use of objective measurements—such as the passive monitoring of usage time and physiological responses, combined with the inclusion of mediating variables such as stress levels, sleep quality, and social skills—could enhance our understanding of the mechanisms underlying the relationship between digital device use and sensory processes. This integrated approach would help clarify the role of smartphones in sensory self-regulation processes, paving the way for targeted interventions to manage digital device use in response to the specific needs of adolescents with ADHD.

## 5. Conclusions

The current evidence suggests important clinical and educational implications, particularly for adolescents with ADHD and their interactions with smartphone use and sensory processing. While intensive smartphone use is associated with increased sensory processing difficulties in adolescents with typical development, this correlation is not observed in adolescents with ADHD. This may indicate that individuals with ADHD already have a higher baseline of sensory processing, which may leave less room for significant changes as a result of smartphone use. Rather than interpreting this lack of correlation as evidence of a regulatory role for smartphones in ADHD, it is important to consider that the high baseline levels of sensory processing in individuals with ADHD may limit the potential effects of smartphone use.

Thus, a more nuanced approach to smartphone use and sensory processing in ADHD is necessary. While controlled smartphone use may offer benefits in some cases, such as managing sensory overload, this relationship may not be universal or as straightforward as previously assumed. It is crucial to adopt individualized strategies that consider the unique sensory processing profiles of adolescents with ADHD.

In educational and therapeutic contexts, these findings suggest that rather than focusing solely on the amount of screen time, attention should be given to how smartphone use interacts with individual sensory processing profiles. Mindful use, tailored to the sensory needs of each adolescent, could potentially offer a more effective approach. Overall, these results highlight the importance of considering multiple factors when integrating digital technologies into the daily lives of adolescents with ADHD, without overemphasizing their role in self-regulation processes.

## Figures and Tables

**Figure 1 ijerph-21-01705-f001:**
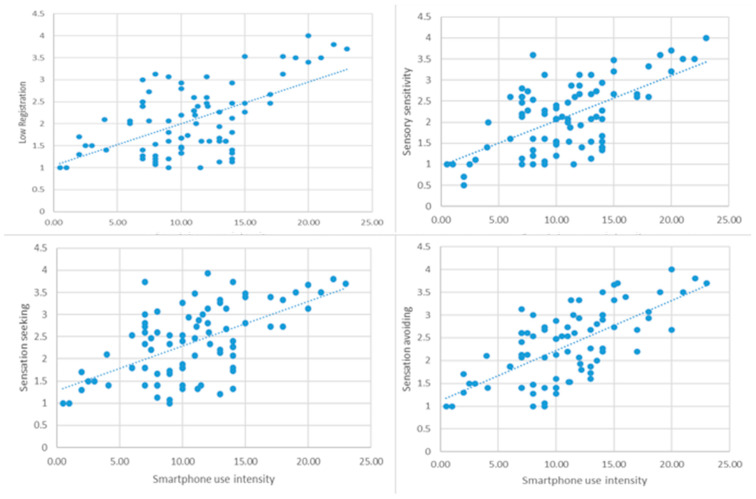
Correlation between smartphone use intensity and the four components of sensory processing in typical development group.

**Figure 2 ijerph-21-01705-f002:**
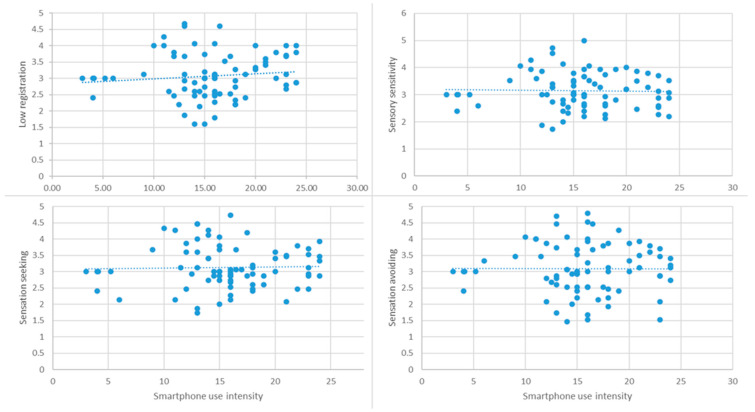
Correlation between smartphone use intensity and the four components of sensory processing in ADHD group.

**Table 1 ijerph-21-01705-t001:** Schematization of the Dunn’s model.

Neurological Threshold	Passive: Acceptance of Stimuli	Active: Management of Stimuli
Low (Hyper-reactivity)	Sensory Sensitivity: Highly responsive to even minimal stimuli.	Sensory Avoidance: Actively avoids intense stimuli to prevent overload.
High (Hypo-reactivity)	Low Registration: Disinterest or distraction, often missing stimuli.	Sensation Seeking: Actively seeks out intense sensory experiences to engage the senses.

**Table 2 ijerph-21-01705-t002:** Demographic characteristics of the sample.

	ADHD	Typical Development
N	92 (57m, 35f)	92 (57m, 35f)
Age (years)	16.61 (±1.82)	16.58 (±1.69)
IQ	108.97 (±10.02)	110.35(±9.47)
Inattention	28.02 (±3.54)	3.88 (±3.61)
Hyperactivity	25.38 (±5.13)	3.41 (±3.20)

**Table 3 ijerph-21-01705-t003:** Smartphone use questionnaire.

1. How much time per day do you spend using smartphone *	☐1 ☐2 ☐3 ☐4
2. I feel lost without my smartphone	☐1 ☐2 ☐3 ☐4
3. I spend more time on my smartphone than I intend to	☐1 ☐2 ☐3 ☐4
4. I check my smartphone frequently throughout the day	☐1 ☐2 ☐3 ☐4
5. I use my smartphone during meals or other social events	☐1 ☐2 ☐3 ☐4
6. I have trouble putting my smartphone down, even when I know I should	☐1 ☐2 ☐3 ☐4

* 1 (never–1 h per day), 2 (2–6 h per day), 3 (7–11 h per day), 4 (more than 12–per day).

**Table 4 ijerph-21-01705-t004:** Descriptive statistics of sensory profile components and smartphone use intensity.

	ADHD	Typical Development
Sensory profile components		
Low Registration	2.84 (±0.37)	1.98 (±0.39)
Sensory Sensitivity	3.14 (±0.71)	2.13 (±0.69)
Sensation Seeking	3.13 (±0.69)	2.11 (±0.74)
Sensation Avoiding	3.09 (±0.23)	2.15 (±0.66)
Smartphone use intensity	15.95 (±3.52)	10.88 (±2.88)

## Data Availability

Data are available on request to each of the authors.

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
