# Peer review of "The Role of Smartphone Use in Sensory Processing: Differences Between Adolescents with ADHD and Typical Development"

_ijerph, 2024, doi:10.3390/ijerph21121705_

Round 1

Reviewer 1 Report

Comments and Suggestions for Authors

Thank you very much for inviting me to review this study. I find the research angle to be innovative, as it explores the relationship between smartphone usage intensity and sensory processing, particularly comparing the differences between adolescents with ADHD and typically developing adolescents in this regard. This study indeed holds certain reference value. However, I have some concerns and doubts about the details presented in the manuscript. The following are my concerns:

1. The current sample has limitations in terms of cultural, socio-economic background, and geographical regions. I recommend that the authors expand the samples to include adolescents from diverse backgrounds, which would enhance the generalizability and representativeness of the results.

2. The study employed self-report methods to measure smartphone usage intensity, which may introduce response bias. I suggest that the authors consider to add objective measurement tools (such as passive monitoring apps) to more accurately assess the time and frequency of smartphone use. The validity and reliability of these tools in ADHD research should also be further explored.

3. The manuscript does not differentiate between specific types of smartphone usage (e.g., for homework versus entertainment). I recommend that the authors consider this variable in future research, exploring the different impacts of various types of smartphone use on sensory processing in adolescents with ADHD.

Comments on the Quality of English Language

The language can be improved to power the research.

Author Response

Dear Editor,

below, we respond to your comments and those of the reviewers, addressing each point individually.

Reviewers' comments:

Reviewer 1

Thank you very much for inviting me to review this study. I find the research angle to be innovative, as it explores the relationship between smartphone usage intensity and sensory processing, particularly comparing the differences between adolescents with ADHD and typically developing adolescents in this regard. This study indeed holds certain reference value.

Reply

Thank you.

However, I have some concerns and doubts about the details presented in the manuscript. The following are my concerns:

  1. The current sample has limitations in terms of cultural, socio-economic background, and geographical regions. I recommend that the authors expand the samples to include adolescents from diverse backgrounds, which would enhance the generalizability and representativeness of the results.

Reply

Thank you. Our study focused on a specific population—Italian adolescents attending public secondary schools in Sicily. While this choice provided a well-defined and homogeneous sample for the initial exploration of our research questions, we acknowledged that it limited the generalizability of the findings to adolescents from other cultural, socio-economic, or geographical contexts.

Expanding the sample to include adolescents from diverse backgrounds would have enhanced the representativeness and broader applicability of our results. However, due to logistical and resource constraints, we chose to maintain internal validity by controlling cultural and regional variability within this study.

We have addressed this limitation in the "Limitations" section of the manuscript and suggested that future research replicate the study with more diverse samples to test the robustness and generalizability of our findings.

  1. The study employed self-report methods to measure smartphone usage intensity, which may introduce response bias. I suggest that the authors consider adding objective measurement tools (such as passive monitoring apps) to more accurately assess the time and frequency of smartphone use. The validity and reliability of these tools in ADHD research should also be further explored.

Reply

Thank you. We appreciated the reviewer’s comment regarding the use of self-report methods to measure smartphone usage intensity. While the questionnaire we employed was adapted from validated measures in the literature and demonstrated good internal consistency (Cronbach’s α = 0.87), we acknowledge that self-report methods are subject to response bias. Incorporating objective tools, such as passive monitoring apps, would have allowed for a more precise evaluation of smartphone usage in terms of time and frequency. We have acknowledged this point as a limitation in the "Limitations" section of the manuscript. Furthermore, we agree with the reviewer’s suggestion that future studies should explore the validity and reliability of objective tools in ADHD research, as they have the potential to complement self-reported data and provide a more comprehensive understanding of smartphone usage.

  1. The manuscript does not differentiate between specific types of smartphone usage (e.g., for homework versus entertainment). I recommend that the authors consider this variable in future research, exploring the different impacts of various types of smartphone use on sensory processing in adolescents with ADHD.

Reply

Thank you. While our study focused on the overall intensity of smartphone use, we recognize that the purpose of usage (e.g., for homework versus entertainment) may have distinct impacts on sensory processing in adolescents with ADHD. Differentiating between these types of usage could provide a more nuanced understanding of their effects. We have addressed this point as a limitation in the "Limitations" section of the manuscript and suggested that future research explore the potential differential impacts of various types of smartphone use. This addition underscores the need for more granular investigations in this area.

The language can be improved to power the research.

Reply

Thank you. We have carefully reviewed and revised the language throughout the manuscript to improve clarity and readability. We hope these improvements enhance the overall quality and impact of the research presentation.

We hope these revisions address your concerns and provide a clearer, more nuanced interpretation of the results. Thank you again for your valuable feedback, and we look forward to your further thoughts on the revised manuscript.

Reviewer 2 Report

Comments and Suggestions for Authors

Reading and interpreting this paper requires a lot of theoretical understanding of the vocabulary of sensory processing.  It is not a paper for physicians who treat ADHD.

There is an introductory general explanation that outlines the descriptive theory of sensory processing.

I would find this a great deal clearer in a grid, as I have drawn below  – but I am not sure that the grid actually makes sense, even assuming I have drawn it correctly from the description. For example, is disinterest/distraction passive acceptance or active management? Is sensory sensitivity passive acceptance or active management? Which of sensation seeking and sensory avoidance is active and which is passive?

Neurological threshold

Passive response: acceptance of stimuli

Active response: management of stimuli

Low (hyper-reactivity)

Low registration

Sensory sensitivity

High (hypo-reactivity)

Sensory avoidance

Sensation seeking

This is made all the more confusing  by the contradiction between line 54: Sensory avoidance pertains to those who, despite having a high threshold, avoid intense stimuli to prevent overload. and line 202: while Sensation Avoiding is the quadrant representing low thresholds with an active response.

The grid as described in 2.3.3 looks different.

Neurological threshold

Passive response: acceptance of stimuli

Active response: management of stimuli

Low (hyper-reactivity)

Sensory sensitivity

Sensory avoidance

High (hypo-reactivity)

Low registration

Sensation seeking

Each person then has a grid for each of the 5 (or 6) sensory modalities, which could be different across modalities. This gives a total of 24 sensory descriptors. In order to evaluate deviations from normality, reference data for each of these are required.

‘Individuals with ADHD tend to exhibit more intense and less adaptive responses to sensory stimuli, likely due to a reduced ability to filter sensory information and a less efficient self-regulation system.’

Surely the sensory threshold will fluctuate according to a person’s engagement in a task, with sensory sensitivity to distractions increasing as the concentration wanes.

Regarding smartphone use, what are the difficulties in sensory processing associated with intensive use? What do they actually look like in relation to specific everyday functions?

What do the sensory profile components look like? Specific examples would be helpful. Are these simply a different vocabulary for describing difference in attention? Eg some sensations can help stimulate the attention and assist with task completion (eg background music, which might have to be of a specific genre) and others are annoying distractions to a person trying to concentrate. Stimulating/interesting/rewarding activities may maintain the concentration.

Regarding the data collected, the ADHD assessment, cognitive testing and smartphone use questionnaire appear to be straightforward. However, to find out about the specific detail of the Adolescent Sensory Profile, a registration fee is required. This does not help with the understanding of what it means. However, it has 60 questions items, for 24 components and modalities with responses on a 5-point scale. I was able to find a few examples of questions, eg: ’I am distracted if there is a lot of noise around’ and ‘I am unsure of footing when walking on stairs’. These appear very general. The first would have to be task-specific and the second might be affected by the lighting and whether the mind is actively engaged in another activity.

It appears that in the analysis the sensory profiles for the different modalities were combined, although this is not entirely clear.

Those with ADHD scored higher in all aspects of sensory profile and had higher smart phone use, but there was no correlation between these.

For those without ADHD there was a correlation between sensory profile and smart phone use. The authors appear to infer causation ‘suggesting that prolonged smartphone use may contribute to sensory overload, resulting in challenges in stimulus regulation.’ The authors have suggested that in the non-ADHD smart phone use may be detrimental, but that in ADHD it may assist with self-regulation. It should also be discussed that in people without ADHD, smart phone use might be assisting with self-regulation among those who already have higher sensory profile scores.

The conclusion – of ‘smartphone use and its potential role in sensory and emotional self-regulation processes.’ is surely going too far and over-interpreting the lack of correlation between smartphone use and sensory processing in ADHD. Other interpretations should be considered, for example, that people with ADHD start from a higher baseline with regard to sensory processing, with less scope for change associated with smartphone use.  

Author Response

Dear Editor,

below, we respond to your comments and those of the reviewers, addressing each point individually.

Reviewers' comments:

Reviewer 2

Reading and interpreting this paper requires a lot of theoretical understanding of the vocabulary of sensory processing.  It is not a paper for physicians who treat ADHD.

There is an introductory general explanation that outlines the descriptive theory of sensory processing.

I would find this a great deal clearer in a grid, as I have drawn below  – but I am not sure that the grid actually makes sense, even assuming I have drawn it correctly from the description. For example, is disinterest/distraction passive acceptance or active management? Is sensory sensitivity passive acceptance or active management? Which of sensation seeking and sensory avoidance is active and which is passive?

Neurological threshold

Passive response: acceptance of stimuli

Active response: management of stimuli

Low (hyper-reactivity)

Low registration

Sensory sensitivity

High (hypo-reactivity)

Sensory avoidance

Sensation seeking

This is made all the more confusing  by the contradiction between line 54: Sensory avoidance pertains to those who, despite having a high threshold, avoid intense stimuli to prevent overload. and line 202: while Sensation Avoiding is the quadrant representing low thresholds with an active response.

The grid as described in 2.3.3 looks different.

Neurological threshold

Passive response: acceptance of stimuli

Active response: management of stimuli

Low (hyper-reactivity)

Sensory sensitivity

Sensory avoidance

High (hypo-reactivity)

Low registration

Sensation seeking

Each person then has a grid for each of the 5 (or 6) sensory modalities, which could be different across modalities. This gives a total of 24 sensory descriptors. In order to evaluate deviations from normality, reference data for each of these are required.

‘Individuals with ADHD tend to exhibit more intense and less adaptive responses to sensory stimuli, likely due to a reduced ability to filter sensory information and a less efficient self-regulation system.’

Surely the sensory threshold will fluctuate according to a person’s engagement in a task, with sensory sensitivity to distractions increasing as the concentration wanes.

Regarding smartphone use, what are the difficulties in sensory processing associated with intensive use? What do they actually look like in relation to specific everyday functions?

What do the sensory profile components look like? Specific examples would be helpful. Are these simply a different vocabulary for describing difference in attention? Eg some sensations can help stimulate the attention and assist with task completion (eg background music, which might have to be of a specific genre) and others are annoying distractions to a person trying to concentrate. Stimulating/interesting/rewarding activities may maintain the concentration.

Regarding the data collected, the ADHD assessment, cognitive testing and smartphone use questionnaire appear to be straightforward. However, to find out about the specific detail of the Adolescent Sensory Profile, a registration fee is required. This does not help with the understanding of what it means. However, it has 60 questions items, for 24 components and modalities with responses on a 5-point scale. I was able to find a few examples of questions, eg: ’I am distracted if there is a lot of noise around’ and ‘I am unsure of footing when walking on stairs’. These appear very general. The first would have to be task-specific and the second might be affected by the lighting and whether the mind is actively engaged in another activity.

It appears that in the analysis the sensory profiles for the different modalities were combined, although this is not entirely clear.

Those with ADHD scored higher in all aspects of sensory profile and had higher smart phone use, but there was no correlation between these.

For those without ADHD there was a correlation between sensory profile and smart phone use. The authors appear to infer causation ‘suggesting that prolonged smartphone use may contribute to sensory overload, resulting in challenges in stimulus regulation.’ The authors have suggested that in the non-ADHD smart phone use may be detrimental, but that in ADHD it may assist with self-regulation. It should also be discussed that in people without ADHD, smart phone use might be assisting with self-regulation among those who already have higher sensory profile scores.

The conclusion – of ‘smartphone use and its potential role in sensory and emotional self-regulation processes.’ is surely going too far and over-interpreting the lack of correlation between smartphone use and sensory processing in ADHD. Other interpretations should be considered, for example, that people with ADHD start from a higher baseline with regard to sensory processing, with less scope for change associated with smartphone use.  

------------------------------------------------------------------------------------------------------------

Reply

Based on your comments, we have made several revisions to the manuscript to clarify the relationship between sensory processing and smartphone use in adolescents with ADHD and those with typical development. Below, we address your concerns, particularly regarding the grid and interpretation of results.

With reference to your first point (Sensory Processing Model and Grid), we have revised the sensory processing grid to better reflect Dunn's model and your suggestions. The grid now clearly distinguishes between the neurological threshold and the responses to stimuli:

Neurological Threshold

Passive: Acceptance of Stimuli

Active: Management of Stimuli

Low (Hyper-reactivity)

Sensory Sensitivity: Highly responsive to even minimal stimuli.

Sensory Avoidance: Actively avoids intense stimuli to prevent overload.

High (Hypo-reactivity)

Low Registration: Disinterest or distraction, often missing stimuli.

Sensation Seeking: Actively seeks out intense sensory experiences to engage the senses.

This revised grid aligns with Dunn’s model, and we hope it clarifies the interaction between sensory thresholds and responses. We also agree with your observation that smartphone use could affect sensory processing differently in individuals with ADHD compared to those with typical development.

With reference to your second point, clarification of Contradiction in the Text, you correctly pointed out a potential contradiction regarding sensory avoidance and sensation seeking. The revised grid should help resolve this by emphasizing that while Sensory Avoidance involves actively avoiding overstimulation (despite having a high threshold), Sensation Seeking refers to the active pursuit of stimulating sensory experiences. We have updated the text accordingly to ensure consistency in the explanation (highlighted in yellow).

With reference to your third point, Specific Examples of Sensory Profiles, we understand the need for more concrete examples of sensory profiles, particularly in relation to real-life situations. For instance:

Sensory Sensitivity: Adolescents with this profile might become distracted by background noise, such as the hum of a fan or the sound of typing, while studying.

Sensation Seeking: These individuals might seek out louder or more stimulating environments, such as listening to music or engaging in physically active tasks, to maintain focus and motivation.

Sensory Avoidance: An adolescent who is sensitive to sensory overload might avoid crowded or noisy spaces to maintain comfort.

Low Registration: An individual in this category may not notice important sensory stimuli, such as a ringing phone or people talking, even though these stimuli are noticeable to others.

We added it and highlighted in yellow.

In response to your fourth request for more detailed discussion on how smartphone use influences sensory processing, we have revised the manuscript to reflect the nuanced findings of our study.

Our analysis revealed that adolescents with ADHD exhibited significantly higher scores across all sensory processing components compared to those with typical development (Table 3). Furthermore, adolescents with ADHD reported significantly higher levels of smartphone use intensity than their typically developing peers, a finding that aligns with Hypothesis 2 (p < .001).

However, when we examined the relationship between smartphone use intensity and sensory processing, distinct patterns emerged. Among adolescents with typical development, we observed a significant positive correlation between smartphone use intensity and sensory processing across all sensory components (r = .370 to r = .443), indicating that increased smartphone use was associated with heightened sensory processing difficulties (Figure 1). This supports the notion that intensive smartphone use may exacerbate sensory overload for adolescents without ADHD.

In contrast, for adolescents with ADHD, the correlation between smartphone use and sensory processing was not significant (r = -.230 to r = .055), suggesting that smartphone use did not have the same impact on sensory processing as it did in the typical development group (Figure 2). This finding raises the possibility that smartphone use may function differently for individuals with ADHD. Specifically, it may not exacerbate sensory processing difficulties in this group, and could potentially even serve as a tool for sensory self-regulation, as suggested by sensory modulation theory.

We acknowledge your suggestion that individuals with ADHD may start with a higher baseline of sensory processing difficulties, which could limit the potential for change with smartphone use. We have amended the manuscript to reflect this interpretation, emphasizing that while smartphone use did not correlate with sensory processing in the ADHD group, it may still play a role in regulating sensory input within this population (highlighted in yellow). For individuals with ADHD, smartphones might offer both stimulation and modulation, potentially helping them manage sensory overload in certain contexts.

For adolescents with typical development, however, the significant correlation between smartphone use and sensory processing suggests that increased use may lead to greater difficulties in regulating sensory input. This difference underscores the importance of considering developmental and clinical profiles when assessing the impact of smartphone use on sensory processing.

Finally, we have revised the conclusion to take into account the complexity of the relationship between smartphone use and sensory processing. Rather than over-interpreting the lack of correlation in the ADHD group, we now emphasize the potential regulatory role of smartphones for this population, while also acknowledging that smartphone use may have detrimental effects on sensory processing in adolescents without ADHD.

We hope these revisions address your concerns and provide a clearer, more nuanced interpretation of the results. Thank you again for your valuable feedback, and we look forward to your further thoughts on the revised manuscript.

Round 2

Reviewer 1 Report

Comments and Suggestions for Authors

All my concerns have been addressed. The language should be improved.

Comments on the Quality of English Language

The language should be improved.